# An Observational Study Using Ultrasound to Assess Allowable Needle Insertion Range of Acupoint CV12

**DOI:** 10.3390/healthcare10091707

**Published:** 2022-09-06

**Authors:** Hongmin Chu, Jaehyo Kim, Seongjun Park, Jaehyun Kim, Jung-Han Lee, Won-Bae Ha, Hyun-Jong Jung, Seung-bum Yang, Cheol-hyun Kim, Jun Yong Park, Kyung-ho Kang, Sangkwan Lee, Sanghun Lee

**Affiliations:** 1Department of Internal Medicine and Neuroscience, College of Korean Medicine, Wonkwang University, Iksan 54538, Korea; 2Department of Internal Medicine, Wonkwang University Gwangju Medical Center, 1140-23 Hoe-jae ro, Namgu, Gwangju 61729, Korea; 3Department of Meridian & Acupoint, College of Korean Medicine, Wonkwang University, Iksan 54538, Korea; 4Department of Medical History, College of Korean Medicine, Wonkwang University, Iksan 54538, Korea; 5Department of Rehabilitation Medicine of Korean Medicine, College of Korean Medicine, Wonkwang University, Iksan 54538, Korea; 6Department of Diagnostics, College of Korean Medicine, Wonkwang University, Iksan 54538, Korea; 7Department of Medical Non-Commissioned Officer, Wonkwang Health Science, Iksan 54538, Korea; 8Department of Sport and Leisure Studies, Yonsei University, Seoul 03722, Korea; 9Yangjae Cheongwoo Korean Medicine Clinic, Seoul 06267, Korea; 10Future Medicine Division, Korea Institute of Oriental Medicine, 1672 Yuseong daero, Yuseong gu, Daejeon 34054, Korea; 11Department of Korean Medicine Life Science, University of Science and Technology, Daejeon 34113, Korea

**Keywords:** safe needling depth, acupuncture, CV12, anthropometric measurements, ultrasonography, adverse events

## Abstract

Background: Abdominal organ injuries are a rarely reported complication when deep acupuncture needling is applied to the abdomen. In order to ascertain the allowable needle insertion range (ANIR) of the abdomen region, we selected acupoint CV12, which is commonly used for treating gastric disease, and ANIR was measured with an ultrasound device. Method: Eighty-five healthy volunteers were recruited, of which 83 cases of ultrasound images were obtained. To investigate the prediction factor of ANIR, we also measured several anthropometric factors. Results: The average ANIR was 25.3 ± 10.2; generally, the ANIRs of females were thicker than those of males; and the liver was observed in 62.7% subjects’ ultrasound images. The non-observed group showed thicker ANIR and higher BMI than the liver-observed group. Conclusion: There are reliable variables that make it possible to predict the ANIR. It is advised to refer to anthropometric factors in needling acupoint CV12 to avoid complications with the acupuncture treatment. However, individual differences are not negligible when applying deep needling. Thus, if the risk is not judged before or during the procedure, practitioners could consider the patient’s ANIR on CV12 when applying acupuncture by examining the individual anatomical structures using ultrasound and considering internal organ positions to prevent adverse events due to acupuncture.

## 1. Introduction

Acupuncture is widely used for treating various diseases [1] and considered a relatively safe intervention [2,3]. Acupuncture is also used to treat gastrointestinal disorders, such as functional dyspepsia [4] and gastroparesis [5]. However, according to the clinical experience or qualification of acupuncturists, serious acupuncture-related complications have occasionally been reported, including irreversible damage to organs and death.

According to several studies of acupuncture-related adverse events, acupoints such as Tianshu (ST25), Zhongwan (CV12), and Qimen (LR14) have accounted for abdominal organ and tissue injuries [6]. In particular, Zhongwan (CV12), the acupoint of the Conception Vessel (CV) located on the upper abdomen’s anterior median line, was the most frequently related to complications following acupuncture treatment [6]. In Zhang’s review, it was confirmed that CV12 was used in 20% of 10 studies with a complication report of abdominal organ damage [6]. The most common complications caused by improper acupuncture needle insertion at CV12 were peritonitis and organ damage, following perforation peritonitis [7,8]. Ancient acupuncture writings also stated that physicians should take great care when apply acupuncture needle at CV12. For example, Huh-Im who wrote Chim-gu-kyung-heom-bang in 1644 stated in his book that doctors should consider patient’s body type when applying acupuncture needle at CV12. Rather than applying 8 cun (fen) depth en bloc, he argued that doctors should change the depth of the injection in accord with the patients’ thickness of outer skin or peritoneum [9]. DongUi-BoGam also mentioned that the needling depth of CV12 is 2.0 cm (8 fen: the conversion of measurement according to Kim’s research [10]).

Although CV12 is frequently selected to treat various diseases and symptoms, if needle insertion is improper, it can cause internal organ damage. Some studies have attempted to determine the safe needling depth or the risk depth of CV12. However, these studies on the needling range or studies of the anatomical structure of CV12 have only been conducted using CT analysis or measure younger aged subjects. Hence, we chose to use an ultrasound device to measure the allowable needle insertion range (ANIR) of acupoint CV12 and conducted an analysis on the depth of CV12 and measured several anthropometric factors to find out if there are any anthropometric factors that could predict organ perforation or damages.

## 2. Materials and Methods

### 2.1. Participants

Studies were conducted between 16 and 26 November 2015 and 1 August and 13 September 2016 at the College of Korean Medicine, Wonkwang University, Iksan, Korea.

Subjects were recruited by posters at Wonkwang University, Wonkwang Health Science University, and other Iksan city areas. All participants provided written informed consent, and these studies obtained approval from the Institutional Review Board (IRB) of Wonkwang University in Iksan, Korea (WKIRB-201510-BM-001 and WKIRB-201606-SB-033). These clinical studies are also registered at the Clinical Research Information Service (CRIS; Osong (Chungcheongbuk-do): Korea Centers for Disease Control and Prevention, Ministry of Health and Welfare (Korea) KCT0002498 and KCT0002506. All methods utilized in these clinical trials were performed in accordance with the STROBE guideline and regulations [11].

One licensed Korean Medicine doctor and one sixth-year Korean Medicine school student palpated the acupoint and took ultrasound images of CV12 under the guidance of a meridian and acupoint professor. Researchers also measured the subject’s anthropometric variables, such as neck circumference (NC), shoulder width (SW), abdominal circumference (AC), etc.

The first study (KCT0002498), conducted in 2015, obtained the ultrasound images of the acupuncture points at the chest, abdomen, and neck from 30 subjects (15 male and 15 female). From those subjects, the images of acupoints LU1, CV14, GB25, CV4, SP11, CV15, SP14, CV23, ST9, GB24, LR13, CV12, CV22, CV20, and ST25 were taken.

The second study (KCT0002506), conducted in 2016, obtained the ultrasound images of the acupuncture points at the face, neck, back, and buttock from 55 subjects (25 male and 30 female subjects). From those subjects, eight high risk acupoints in the face and neck region (ST6, ST7, SI19, TE17, GV15, GV16, GB20, and GV14), eleven acupoints at the shoulder and thoracic region (GB21, SI14, BL12, GV12, BL13, BL15, GV9, BL17, BL46, BL21, and BL50), eleven acupoints at the lumbar and pelvic areas (GV4, BL23, BL52, GV3, BL25, BL31, BL32, BL33, BL34, and GB30), and CV12 were measured. Only the acupoint CV12 was taken in common in both clinical trials.

Since this was an observational registry study, a sample size calculation was not required. However, researchers planned to set the number of participants at a minimum of 80.

#### 2.1.1. Inclusion Criteria

All subjects were aged between 18 to 39 and were recruited through posters put up at Wonkwang University, Wonkwang Health Science University, and other Iksan areas.

Participants agreed to participate and signed the consent form after hearing a clear explanation of the purpose and characteristics of this clinical study.

#### 2.1.2. Exclusion Criteria

(1) Participants who were pregnant or nursing mothers.

(2) Participants who were diagnosed with liver, kidney, nervous system, immune system, respiratory, endocrine, cardiovascular, or circulatory diseases; tumors; or mental illness.

(3) Subjects with a history of surgery or skin disease at the site of ultrasound imaging.

(4) Abnormal shadows, such as tumors, found at the site of ultrasound imaging.

(5) Those judged to be inadequate for clinical research or have any other reason deemed inappropriate for participation in research by the medical staff.

### 2.2. Location of Acupoint CV12

We palpated acupoint CV12, which is at the midpoint of the line connecting the end of xiphoid process and the center of the umbilicus, according to the WHO Standard Acupuncture Point Locations in the Western Pacific Region (Figure 1). Subjects were in the supine position when obtaining an ultrasound image of CV12 [12]. Palpitation procedure of CV12 methods are shown in Figure 2. The procedure to acquire ultrasound images of CV12 followed the same steps described in our previous study [13].

### 2.3. Acquiring Ultrasound Imaging of Acupoint CV12

CV12 is located at the midpoint of the line connecting the xiphisternal junction and the center of umbilicus. It is on the anterior median line of the upper abdomen, at the 4 B-cun superior to the center of the umbilicus. We used an ultrasound machine, GE voluson 730 expert (General Electric Co., Boston, MA, USA), and linear ultrasound probes (2D linear array Transducers, General Electric Co., Boston, MA, USA) to acquire the ultrasound images of acupoint CV12. Convex probes (2D curved array transducers, General Electric Co., Boston, MA, USA) were used to additionally check whether there were any abnormalities on the liver when the liver was not seen by the linear probe. To determine the risk of liver damage in applying acupuncture on CV12, linear probe imaging was initially used to determine the position of the liver. If the liver was unobserved due to severe abdominal obesity, convex probe imaging was used. The probe’s placement for ultrasound is shown in Figure 2.

### 2.4. Analysis and Measurement

The measurement of the maximal safe needling depth of CV12 was defined as the distance between the skin and visceral peritoneum, which could be measured using ultrasound imaging. The probe angle was vertical to the skin over CV12 and parallel to the sagittal plane. The distance was measured using the scale mounted on the ultrasound device. The ultrasound image of CV12 is shown in Figure 3A,B. The researchers identified the cases in which the liver was located at the acupoint CV12 in the acquired images to determine the risk of damage when performing acupuncture treatment on CV12. We defined ANIR from the surface of the skin to peritoneum and measured it with the ultrasound.

### 2.5. Measurement of Anthropometric Parameters

The participants’ weight and height were automatically measured using an InBody device (Inbody BMS330; Biospace, Co., Seoul, Korea).

Body weight was calculated to the nearest 0.1 kg, and standing height was measured to the nearest 0.1 cm. BMI was calculated body weight (kg) divided by height (m) squared.

Anthropometric measurements were performed with reference to the Anthropometry Procedures Manual 2016 of the National Health and Nutrition Examination Survey [14]. The values of anthropometric factors were measured using measurement tapes (Mr. Home, China). In recruiting subjects and analyzing the results, researchers classified the subjects based on the international standard BMI scale, instead of the new standard BMI adjusted to Korean [15], because this new criteria is not widely used yet.

### 2.6. Satistical Analysis

Statistical analysis was performed using R version 1.0.143 software (R Studio, Boston, MA, USA). Categorical variables were expressed as the numbers and percentages of patients, whereas continuous variables were expressed as means (standard deviations) or medians (quartiles). The correlation between the depth of the CV12 and the anthropometric indicators was calculated using Pearson’s correlation coefficient. A *p*-value < 0.05 was considered significant. Student’s t-test was used to compare the characteristics of the groups whose liver is located at the point of CV12 and those without liver on CV12.

## 3. Results

### 3.1. Subject Characteristics

Out of 85 subjects, we statistically analyzed 83 subjects. Two subjects were excluded (in one, we did not obtain clear ultrasound images, and in one, we did not obtain ultrasound images because of severe obesity). Among the 83 subjects, 40 were male and 43 were female, and the average age of males was 20.4 ± 2.6 years and females were 20.3 ± 2.4 years. The average age of the total subjects was 20.3 ± 2.5 years. The subjects’ total average BMI was 22.4 kg/m^2^. The average BMI of the male subjects was 23.0 ± 4.6 kg/m^2^, and that of the female subjects was 21.9 ± 3.7 kg/m^2^.

Based on the BMI standards, according to WHO criteria [16], below 18.5 kg/m^2^ is underweight, 18.5 kg/m^2^ to 24.9 kg/m^2^ is normal weight, and 25 kg/m^2^ or higher is overweight. Based on the BMI standards out of 83 subjects, 14 were underweight, 47 were normal, and 22 were overweight. The characteristics of the subjects and values of the BMI groups and their demographic characteristics are shown in Table 1. Representative ultrasound images of CV12 of the underweight, normal weight, and overweight group observed through the linear probe are shown in Figure 3C. The flowchart of whole study is shown in Figure 4.

### 3.2. Safe Needling Depth of CV12

Ultrasonography was used to measure the subcutaneous fat thickness and the depth from the skin to the peritoneum. The average depth of male subjects’ subcutaneous fat was 13.4 mm at acupoint CV12, and the female subjects’ was 18.5 mm. The average depth of male subjects’ peritoneum was 22.3 mm at acupoint CV12, and the female subjects’ was 28.1 mm. Table 2 indicates the average thickness of subcutaneous fat at acupoint CV12 and the average depth of skin to peritoneum at acupoint CV12

### 3.3. The Degree of Risk of Liver Damage in CV12 and Physical Characteristics of the Risk Group

Out of the total 83 subjects, 52 subjects’ livers were observed under the peritoneum and 31 subjects’ livers were not observed under the peritoneum. In the latter, researchers observed the stomach rather than the liver. The BMI of those groups whose liver was visible was 21.6 ± 3.9 kg/m^2^, with the depth to visceral peritoneum being 23.2 ± 10.5 mm. The BMI of those groups whose liver was not visible was 23.8 ± 4.3 kg/m^2^, with the depth to visceral peritoneum being 28.8 ± 8.6 mm. There was a statistically significant difference between the two groups in terms of BMI (*p* < 0.01) and the depth of peritoneum (*p* < 0.01). Additionally, there was a statistically significant difference between male and female subjects in depth to the peritoneum (*p* < 0.05). On average, the female subjects had a deeper depth. The corresponding content of boxplot graphs can be found in Figure 5.

### 3.4. Correlation between Needling Depth and Anthropometric Parameters in the Study Group

The correlation coefficients between the independent and dependent variables used in the regression analysis are shown in Table 3. ANIR was significantly positively correlated with AC (r = 0.620, *p* < 0.01), HC (r = 0.507, *p* < 0.01), and BMI (r = 0.662, *p* < 0.01). NC is SW (r = 0.385, *p* < 0.01), AC (r = 0.478, *p* < 0.01), HC (r = 0.309, *p* < 0.01), HT (r = 0.325, *p* < 0.01), and BMI (r = 0.409, *p* < 0.01). SW was positively correlated with AC (r = 0.299, *p* < 0.01), HC (r = 0.271, *p* < 0.05), and HT (r = 0.249, *p* < 0.05). HC had a positive correlation with BMI (r = 0.656, *p* < 0.01) and Uxp with height (r = 0.256, *p* < 0.05). Usp did not show any correlation with any variable. The scatter plots of the measured values are shown in Figure 6.

### 3.5. Regression Analysis Results for ANIR Related to Anthropometric Parameters

In order to examine the variables affecting ANIR, a stepwise linear regression analysis was performed using the abdomen, hip, and BMI, which were significantly correlated, as independent variables. As a result, BMI was the only significant explanation for ANIR (β = 0.662, t = 7.950, *p* < 0.001) (Table 4). This result can be interpreted that the larger the BMI value, the deeper the ANIR. However, since BMI is a value calculated through body measurement indicators using weight and height, multiple regression analysis was re-conducted by inputting actual body measurement variables. Abdomen circumference and hip circumference were values significantly correlated with ANIR. As a multiple stepwise linear regression result, only the abdomen had a significant relationship with ANIR (β = 0.620, t = 7.114, *p* < 0.001). Multiple regression analysis results are shown in Table 4.

Furthermore, we conducted the multiple regression analysis of the anthropometric factors of the group where the liver was observed and only BMI showed a significant relationship with ANIR.

### 3.6. Sub-Group Analysis

There was a significant difference in BMI between the group where the liver was observed and the group where it was not observed. (*p* < 0.05) In addition, a significant difference was also seen in the Uxp (Xiphoid process to umbilicus) between the two groups (*p* < 0.05). Further analysis was conducted on the height to determine the covariate, because we hypothesized that as variables ‘heights’ were related to variable ‘Uxp’. However, no significant difference was observed between the two groups. (*p* = 0.11)

## 4. Discussion

In this observational trials of 85 young, healthy volunteers’ CV12 acupoint using ultrasonography, the ANIR of CV12 was observed less than 3 cm and the standard deviation differed greatly. Furthermore, a correlation analysis was conducted between ANIR and anthropometric indicators and BMI, AC, and HC were significantly associated with ANIR. BMI was identified as an independent predictor of ANIR in regression analysis (y = −10.960 + 0.662 × BMI).

Moreover, we compared the group characteristic between group subjects where the liver was visible and group subjects where the liver was not visible. The liver is an organ in the abdomen region which is not protected by the skeletal system [17]. Additionally, the liver is susceptible to trauma injury because of inadequate connective tissue, which can bring on severe complications, such as hematoma and peritonitis [18]. Since there is a risk of liver damage during acupuncture, we judged that predicting the ANIR of CV12 is important for medical practitioners performing acupuncture in the clinical field and presented a predictable formula through regression analysis. In the regression analysis on the group where the liver was visible, BMI was identified as an independent predictor of ANIR (y = −12.271 + 0.620 × BMI).

Notably, in the correlation analysis, Uxp did not show significant association. However, Uxp displayed significant association in the subject group where the liver was observed. Uxp is one of the proportional bone measurements that is conventionally used to find acupoint locations accurately. The WHO standard acupuncture location measurement also affirms the use of Uxp to find the location of CV12. It is meaningful that traditional measurement methods are still feasible today. Furthermore, we suggest that if a subject’s BMI is low and the Uxp is short, this study’s results recommend that physicians apply acupuncture treatment on CV12 with caution.

Since this study is an observational study, there are several limitations inherent in the study design. One limitation is that the results of this study cannot be directly applied to other age groups or patient groups because this study was conducted on healthy people in their twenties. If physicians apply the reference range of this study in clinics, they should consider these results do not cover all ages. Therefore, future studies should focus on various variables that can judge ANIR regarding changes in breathing and posture or the differentiation between seniors or adolescents.

This is the first study that investigates liver location at CV12 and examines the correlation between anthropometric factors and their association with ANIR in healthy volunteers in their twenties. While acupoint CV12 is widely used by physicians for treating various gastrointestinal diseases, on the other hand, there is still the risk of undergoing complications, such as organ damage, when applying acupuncture treatment to CV12. This is the reason why we focused on CV12. Additionally, as this study was an observational study conducted in healthy subjects, and there were no adverse events during the study process.

Recent studies have suggested that using ultrasonography may be helpful for avoiding adverse event following acupuncture treatments and educational efficacy [19,20,21,22,23]. In the results of this study, it was also possible to predict the tendency of the ANIR to appear deeper when the BMI of subjects increased, but it was difficult to know whether the liver was damaged when the acupuncture needle was actually inserted. Among the total subjects, 62% indicated that the liver was located under acupoint CV12. This means that if the risk is not judged before or during the procedure, it could be helpful to use ultrasound to avoid complications and perform a safe procedure.

## 5. Conclusions

This study aimed to determine the safe depth of acupoint CV12. According to 83 subjects’ ultrasound images, the average ANIR was 25.29 ± 10.20 mm. In addition, the depth was greater in female subjects than in male. The liver was identified in 52 subjects’ ultrasound images (out of 83 cases). Furthermore, BMI and ANIR of the group where the liver was not visible were greater than those of the group where it was visible. Regarding the correlation between anthropometric factors and ANIR, BMI showed the highest correlation. In terms of regression analysis, BMI and abdominal circumference significantly explained ANIR. Based on this result, it can be inferred that statistical tendencies exist. However, individual differences are not negligible enough to simply apply deep needling, as this could induce severe complications, including organ damage. Thus, if the risk is not judged before or during the procedure, practitioners should consider the patient’s ANIR when applying acupuncture to CV12 by examining the individual anatomical structures using ultrasound and considering internal organ positions to prevent adverse events due to acupuncture. 

## Figures and Tables

**Figure 1 healthcare-10-01707-f001:**
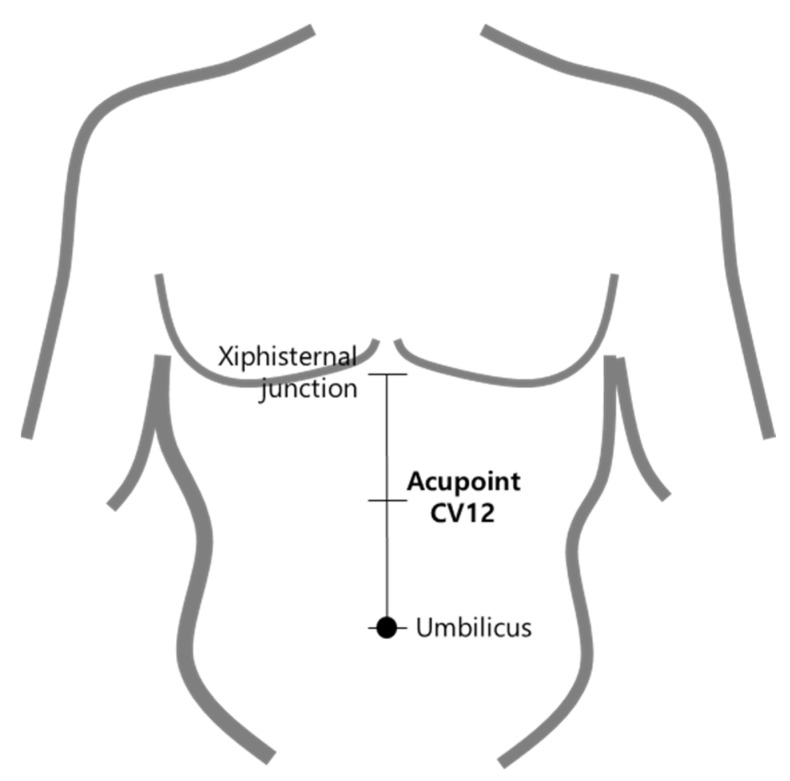
Location of Acupoint CV12.

**Figure 2 healthcare-10-01707-f002:**
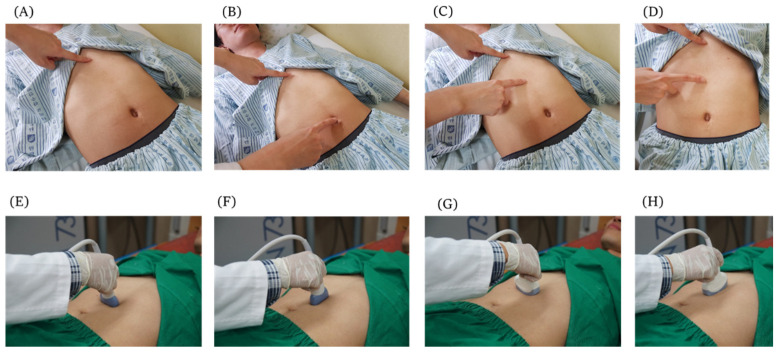
Palpation and probe placement for ultrasound imaging of acupoint CV12 (**A**) Palpitate xiphisternal junction (**B**) Palpitate center of umbilicus (**C**) Palpitate midpoint of the line connecting the end of xiphisternal junction and the center of the umbilicus. (**D**) High angle view of acupoint CV12. During ultrasound imaging, the probe was placed vertically and horizontally on the skin over acupoint CV12, following the needle direction, in order to efficiently measure the distance from the skin to the peritoneum. (**E**) Linear probe placed on the CV12, perpendicular to median line. (**F**) Linear probe placed on the CV12, parallel to median line. (**G**) Convex probe placed on the CV12, perpendicular to median line. (**H**) Convex probe placed on the CV12, parallel to median line.

**Figure 3 healthcare-10-01707-f003:**
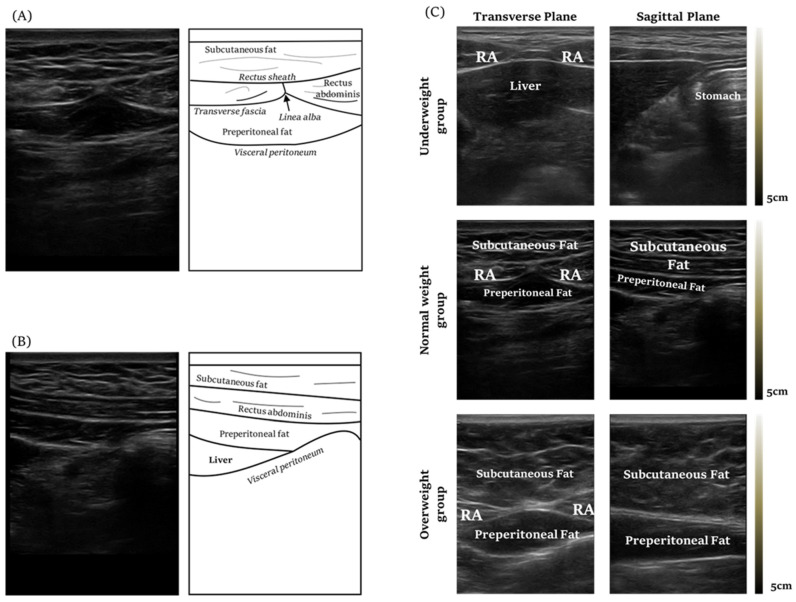
Ultrasound image of CV12 (**A**) Ultrasound image, transverse plane view of CV12. The rectus abdominis is seen beneath the subcutaneous fat. Underneath is the preperitoneal fat. The visceral peritoneum lies below the preperitoneal fat. (**B**) Ultrasound image, coronal view of CV12. (**C**) Ultrasound image of CV12 according to the body mass index.

**Figure 4 healthcare-10-01707-f004:**
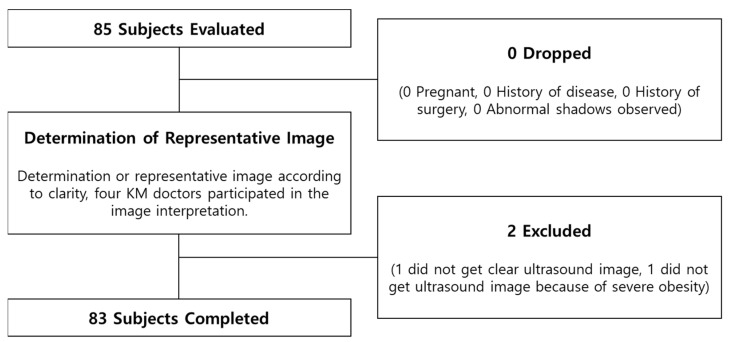
Flowchart of Study.

**Figure 5 healthcare-10-01707-f005:**
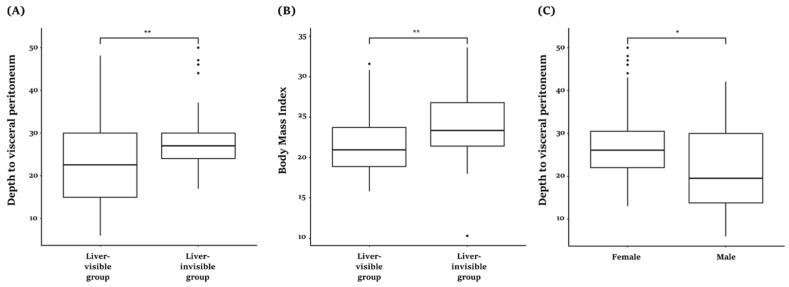
Boxplot graph of acupoint CV12. * *p* < 0.05, ** *p* < 0.01.

**Figure 6 healthcare-10-01707-f006:**
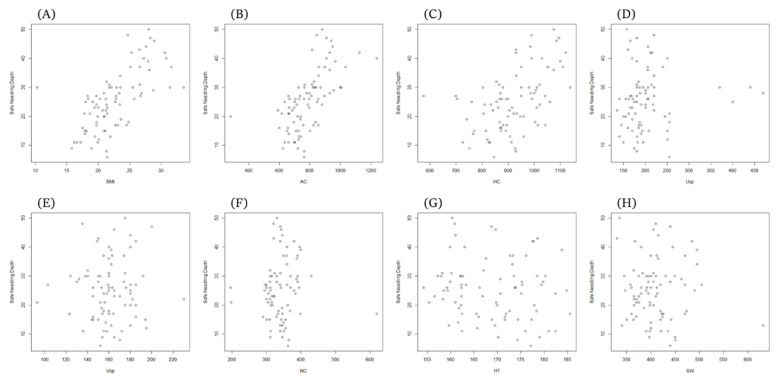
Scatter plot graph showing correlation (r) between needling depth (cm) and various parameters. The scatter plots indicate positive trends in the relationship of all parameters on the safe needling depth. (**A**) BMI and ANIR, (**B**) AC and ANIR, (**C**) HC and ANIR, (**D**) UXp and ANIR, (**E**) USp and ANIR, (**F**) NC and ANIR, (**G**) HT and ANIR, and (**H**) SW and ANIR.

**Table 1 healthcare-10-01707-t001:** Characteristics of the study population.

	Male	Female	Total
Sample	40	43	83
BMI (kg/m^2^)	23.0 ± 4.6	21.9 ± 3.7	22.4 ± 4.2
Underweight (<18.5 kg/m^2^)	7	7	14
Normal Weight (≥18.5 & <25 kg/m^2^)	21	26	47
Overweight (≥25 kg/m^2^)	12	10	22
Neck circumference (mm)	369.4 ± 47.3	313.8 ± 31.0	340.6 ± 48.4
Shoulder width (mm)	438.1 ± 45.6	381.7 ± 26.2	408.9 ± 46.4
Abdomen circumference (mm)	825.4 ± 134.6	748.8 ± 118.8	785.7 ± 132.3
Hip circumference (mm)	928.2 ± 125.6	912.1 ± 88.6	919.8 ± 108.3
Umbilicus to xiphoid process (mm)	216.4 ± 69.6	188.4 ± 37.8	201.9 ± 57.2
Umbilicus to symphysis pubis (mm)	158.1 ± 13.6	162.7 ± 24.9	160.5 ± 20.4

Abbreviations: BMI; Body Mass Index.

**Table 2 healthcare-10-01707-t002:** Depth to subcutaneous fat and peritoneum on acupoint CV12.

		Subcutaneous Fat Thickness at CV12 (mm)	Depth to Peritoneum at CV12 (mm)
Total	Total	16.0 ± 7.7	25.3 ± 10.2
	Underweight	8.1 ± 4.5	15.4 ± 5.9
	Normal Weight	14.0 ± 5.2	22.2 ± 7.5
	Overweight	24.3 ± 6.8	36.9 ± 6.9
Male	Total	13.4 ± 7.5	22.3 ± 10.2
	Underweight	5.9 ± 2.6	12.4 ± 2.4
	Normal Weight	11.4 ± 5.6	18.8 ± 7.5
	Overweight	20.5 ± 5.5	33.9 ± 5.2
Female	Total	18.5 ± 7.1	28.1 ± 9.4
	Underweight	13.3 ± 3.7	23.3 ± 5.8
	Normal Weight	15.8 ± 4.1	24.7 ± 6.4
	Overweight	28.0 ± 6.3	40.1 ± 7.4

Abbreviations: CV12; Acupoint Conception Vessel 12 (Zhong wen).

**Table 3 healthcare-10-01707-t003:** Correlation between depth to peritoneum on CV12 and anthropometric parameters in the study population.

	ANIR	BMI	AC	NC	SW	HC	Uxp	Usp	HT
ANIR									
BMI	0.662 **								
AC	0.620 **	0.788 **							
NC	0.053	0.409 **	0.478 **						
SW	−0.050	0.210	0.299	0.385 **					
HC	0.507 **	0.656 **	0.601	0.309 **	0.271				
Uxp	0.093	0.065	0.273	0.178	0.164	0.157			
Usp	0.042	0.029	−0.073	0.076	0.015	0.091	−0.124		
HT	−0.114	0.101	0.249	0.325 **	0.469	0.181	0.265	−0.001	

** *p* < 0.01. Abbreviations: ANIR, Allowable Needle Insertion Range; AC, Abdominal Circumference; NC, Neck Circumference; SW, Shoulder Width; HC, Hip Circumference; Uxp, Xiphoid process to umbilicus; Usp, Umbilicus to pubic symphysis; HT, Height.

**Table 4 healthcare-10-01707-t004:** Stepwise linear regression and multiple regression analysis results for ANIR in relation to anthropometric parameters.

Stepwise Linear Regression	Variable	B	SE	β	t	F	R^2^
		−10.960	4.638		−2.363 *	63.204 ***	0.431 ***
Independent variable	BMI (kg/m^2^)	1.616	0.203	0.662	7.950 ***		
Multiple Regression	Variable	B	SE	β	t	F	R^2^
		−12.271	5.354		−2.292 *	50.613 ***	0.377 ***
Independent variable	Abdomen (mm)	0.048	0.007	0.620	7.114 ***		

* *p* < 0.05 *** *p* < 0.001. Abbreviations: BMI, Body Mass Index; ANIR, Allowable Needle Insertion Range.

## Data Availability

The datasets used and/or analyzed during the current study are available from the corresponding author on reasonable request.

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
