# Peer review of "An Observational Study Using Ultrasound to Assess Allowable Needle Insertion Range of Acupoint CV12"

_healthcare, 2022, doi:10.3390/healthcare10091707_

Round 1

Reviewer 1 Report

Dear colleagues!

From the point of view of such a science as metrology, measurement results should be rounded up to the third digit. In your case, you should write the following expression 25.3±10.2 mm instead of 25.29±10.20 mm. When writing the fourth digit, the question arises, and by what means of measurement was the measurement accuracy of 0.01 mm achieved? 

The same applies to percentages. Instead of 62.65%, you should write 62.7% according to the rounding rules.

It should not be forgotten that in intermediate statistical calculations it is necessary to work with a large number of digits in order to obtain reliable data on average values, which is observed in this article.

Author Response

Thanks for the detailed reviewer's suggestions.
In accordance with the reviewer's opinion, it has been rounded to the second decimal place and revised whole manuscript.
We will refer your suggestion when writing future research papers.

Thank you!

Reviewer 2 Report

The article has described how safe acupuncture techniques with ultrasound are in their cohort, focusing on acupoint CV12 in high-risk areas. This is an exciting report; however, there are some concerns about this article. 1. Please present each acupuncture point in figures for readers, not specialized. 2. The authors could present the study following diagram as a figure in the article. 3. Is this study any adverse events, including trivial ones? 4. Number of references is small and not updated. 5. Further English editing is required.

Author Response

Thanks for the reviewer`s suggestions. We revised the research paper according to the reviewers' opinions.

First, we add Figure of Acupoint CV12 not specialized format as Figure 1.

Second, we add study flowchart diagram as Figure 3.

We also add adverse events in the discussion section “Also, as this study was an observational study conducted with healthy subjects, there were no adverse events during the study process.”

We add some reference and have revised English throughout whole manuscript.

Thank you!

Reviewer 3 Report

The authors described "An observational study using ultrasound to assess allowable needle insertion range of Acupoint CV12". Recent studies have suggested that using ultrasonography may be helpful for avoiding adverse event following acupuncture treatments. However, these studies on the needling range or studies of anatomical structure of CV12 have only been conducted using CT analysis or measure young-aged subjects. This study measured ANIR with the ultrasound device. So, this topic should be attractive for potential readers. I have some suggestions to improve this manuscript.

1. English editing is needed (e.g. In Abstract, there is duplication of Conclusion and in conclusion).

2. How did you decide the number of subjects? Please add the rationale of eighty-five subjects.

Author Response

Thanks for the kind review.

We have revised English including Abstract and Discussion.

Furthermore, Subjects were recruited twice. In the first clinical data collection study, 30 patients were recruited, and in the second, 55 patients were recruited and 85 patients were recruited. Since this study is an observational study and not a study to test a hypothesis, no statistical methods were applied to the subject recruitment. We supplemented these in 2.1 according to reviewer comments.

Sincerely,